# Back to Exile: Current Jewish Critiques of the Jewish State

**Elad Lapidot**

Le Département Études Romanes, Slaves et Orientales, Faculté des Langues, Cultures et Sociétés Campus Pont-de-Bois, University of Lille, 59653 Villeneuve d'Ascq, France; elad.lapidot@univ-lille.fr

**Abstract:** This article reviews recent books by Jewish thinkers that critique the idea of a Jewish state from the perspective of Jewish exile. It outlines two main approaches. The first, secular approach, rejects the Jewish state in favor of a secular state, seeing Judaism itself as the problem, whether arising from biblical violence or collective identity. The second, post-secular approach, rejects the Jewish state as secular, and finds resources in Jewish tradition for an alternative political vision centered on exile, understood as resistance to sovereignty and violence. This article argues that Jewish opposition to the Jewish state aims to limit sovereignty, integrate Jews into the Middle East space, and recover an exilic Jewish tradition of social ethics and pluralism. The idea of exile thus provides resources for envisioning decolonization and coexistence in Israel–Palestine.

**Keywords:** State of Israel; Jewish state; exile; diaspora; galut; sovereignty; secularism; post-secularism

The 75th anniversary of the State of Israel is no celebration. The difficulties touch the soul of the Jewish state. On the surface, it has been greatly strengthened in recent years. In 2018, a Basic Law was passed declaring that the State of Israel is for the exclusive national existence of the Jewish people. The current government, led for the first time by the national-religious right, has taken drastic steps to strengthen the Jewishness of the state, while causing unprecedented damage to other values, to non-Jewish sectors, and to the stability of the state's institutions themselves.

The shocks were not long in coming. A broad Jewish public opposed the transformation as a threat to the secular and democratic nature of the state. The protest movement swept away the centers of power in Israeli society—the economy, the legal system, and the army and threatened a constitutional crisis and even civil war. But the internal Jewish struggles were abruptly ended by the violence that erupted among Palestinians, from Gaza, in the October 7 Hamas attack.

The horror overwhelmed Israeli Jews, who suddenly felt defenseless. The Jewish state evaporated for a long moment in the minds of many, and the nightmare of persecution raged. Their hearts were pierced by the pain of losing the Jewish state, and the absolute, urgent need for it was felt as a commandment. The feeling of weakness unleashed a lust for power in the killing machines, and in the rage unleashed on Gaza, Israel is revealed as a warlord. In this revelation it becomes clear again that the Jewish state is purchased with human sacrifice.

The political situation in Israel-Palestine, long frozen as a fact, has flared up again as a burning question. At the heart of the situation is the question of the Jewish state. How is violence inherent in the Jewishness of the State of Israel, and how might a conceptual change in the relationship between Judaism and the state change the situation?

The following pages explore this question through a series of writings published in the years leading up to October 7. They are all written by Jewish thinkers who oppose the idea of the Jewish state from a Jewish perspective. All the authors are rooted in Western academic discourse. But they all turn to the Jewish tradition and renew the idea of exile at the heart of contemporary political thought.

They understand the meaning of exile in Judaism in different, contradictory ways. Surprisingly, for most of them exile does not oppose the state; on the contrary, it justifies it.

For those, exilic thought rejects the Jewish state in favor of a non-Jewish state. The others draw from the thought of exile an opposition to the very idea of the state. The disagreement between them stands on the question of secularism, that is, of transcendence.

## 1. Pagan State

A profound criticism of the Jewish state project emerges from the latest book by Israeli expatriate philosopher Adi Ophir, one of the prominent voices of the Israeli left intelligentsia, *In the Beginning Was the State: Divine Violence in the Hebrew Bible* (Ophir 2023). The book does not deal directly with Israel, but I read it—stretching beyond Ophir's explicit intentions in his book—as a sophisticated expression of secular opposition to the Jewish state.

The concern of the book is a critique of the modern state, that is, the model of the state that emerged in modern Europe and that has become the hegemonic model of politics in general. It is part of a broad conversation in contemporary political philosophy with interlocutors such as Walter Benjamin, Michel Foucault, Giorgio Agamben, Jan Assmann, and Michael Walzer. A particularly important interlocutor is Carl Schmitt, since Ophir argues that critical thought must examine the state in its theo-political context, that is, in relation to the idea of God.

Schmitt, the father of political theology, argued that "all basic concepts of modern state theory are secularized theological concepts" (Schmitt [1922] 2009, p. 43). In particular, he pointed to the connection between the concept of the sovereign, which Schmitt argued was the foundation of the modern state, and the concept of God in Western theology. He claimed that modern sovereignty is a secularized embodiment of the monotheistic God as an absolute power beyond any law.

Ophir's fundamental move both reverses and radicalizes Schmitt's claim. Ophir agrees that there is a connection between the power of the sovereign and the power of the monotheistic God who transcends the world order. The reversal is the evaluation. While Schmitt justified the sovereign power of the state, Ophir is critical. His basic concept of sovereign power and the modern state is negative: not just power, but harmful power, "violence".

The radicalization of Schmitt's thesis is at the heart of Ophir's book. Whereas Schmitt argued that the sovereign is the modern substitute for God, Ophir argues that the sovereign is the modern manifestation of God. In other words, if for Schmitt the state is the absence of God, that is, secularized existence, for Ophir the state embodies God: "the state is monotheism's true contemporary figure" (Ophir 2023, p. 252).

The claim is theological: it touches on the nature of the monotheistic God and the nature of His presence. Ophir polemicizes not only with Schmitt, but with the entire monotheistic tradition, including Catholic-Christian, rabbinic-Jewish, and even philosophical, without once mentioning Islam. According to him, all these traditions, from the philosophical God of Philo and Josephus, have presented God as the opposite of violence—as Logos, Spirit, and Law. According to Ophir, this representation is false, which means that God has actually been hidden by tradition, not by modernity, in which God appears in His true form, as sovereign violence. According to Ophir, the tradition of the nonviolent God did not provide an alternative theology. On the contrary, the tradition "repressed" the violent God, that is, it did not eliminate Him, but perpetuated Him.

Resistance to the modern state as sovereign violence requires resistance to the monotheistic God as transcendent violence, and resistance to God requires exposing Him behind the screens of tradition in His true being. Ophir's resistance to the state is thus carried out, and this is the center of the book, through an act of revealing God, resistance through revelation.

Where does God reveal Himself to the philosopher? Ophir seeks revelation before tradition, at its beginning, in its purity. He finds the source in the Bible. The opposition to the modern state is a hermeneutic act: the exposure of "divine violence in the Hebrew Bible". To expose the Jewish God and condemn Him as evil is to repeat the operation of the Gnostic theologians since Marcion, to whom Ophir explicitly refers, who tried to separate Christianity from Old Testament Judaism and was declared a heretic by the Church.

Ophir's polemic with tradition is based on a distinction between two ways of reading Scripture. The traditional reading, according to him, is a concealing, "non-literal" reading that hides the literal meaning of the text in allegories (Christian) and midrashim (rabbinic). Ophir, on the other hand, adopts a reading that seeks to break away from tradition and return to the simple biblical word. He appeals to the doctrine of sola scriptura, which the founder of Protestantism, Martin Luther, turned against the Catholic tradition.

But Ophir challenges not only theological tradition, but also the secular reading of the biblical text. He accuses tradition of covering up God's violence with conciliatory allegories. Secular readers such as Michael Walzer or Jan Assmann have identified and rejected the violence of the biblical God. But, Ophir argues, by clinging to the secularization thesis, they have defined divine violence as "religious" and neutralized its political essence. Ophir's thesis, as noted above, is that the monotheistic God in his biblical, literal manifestation is essentially political violence: "In the beginning was the state."

The bulk of the book is devoted to Ophir's reading of the Pentateuch. It is a fascinating reading, and there is no room here to discuss its details. He presents the Pentateuch as a Hebrew "politeia" in which transcendent divine violence lays the foundation for a political order that Ophir claims will later emerge in the modern state. Ophir identifies three models of Hebrew political theory: three "theo-political formations." Each formation he attributes to one of the historical layers of the Pentateuch text according to the accepted documentary hypothesis: the pre-priestly, the priestly, and the Deuteronomic. The three formations represent three stages in a conceptual-historical development.

The evolution proceeds from violence to law. In the pre-priestly formation, the divine sovereign is revealed as catastrophic violence, without any restraint, as in the Flood in the days of Noah. The priestly model restrains sovereign violence and regulates it through laws of sacrifice. In the last stage, that of the Deuteronomist, violence is sublimated from reality into language, from the present into the past as historical narrative and into the future as threat. Sacrifice is internalized into a rational system of commands of the individual subject. In the internalized divine violence, Ophir sees an ideology of submission to transcendent power, manifested in the ethos of self-sacrifice of the citizen in the modern state.

The alternative vision that Ophir points to is that of a "pagan state" devoid of any transcendent authority, which is secular. The secular state is governed not by divine force but by human law, rational and absolutely immanent, a law without God.

One should ask whether Scripture itself does not offer a vision of the replacement of divine violence with human law. Ophir's readings point to quasi-secular biblical figures such as Joseph, Korah, and Jethro. He claims that this direction is rejected in the text. However, his readings point to a progression from brute violence at the beginning of the Pentateuch to rational individual law at its end.

And Scripture does not stop there. Hermann Cohen, for example, continued to read the development of rationalization from the Pentateuch to the prophets, who influenced the development of a variety of post-biblical traditions. These traditions, philosophical, Christian, and rabbinic, continued intra-biblical trends to develop nonviolent theologies that indeed fulfill Ophir's vision, which he actually recognizes in relation to the anarchic legal cultures of the diasporic rabbinic tradition.

He rejects these traditions not on substance but on hermeneutical grounds, as based on a non-literal reading. Yet Scripture itself, sola scriptura, already suggests complex ways of interpreting itself. The biblical corpus is built on internal intertextuality, one part read by another, the book of Exodus by the book of Deuteronomy, or the Pentateuch by the prophets. This hermeneutic, revealed in revelation itself, has guided later traditions.

Is not the absolute break that Ophir seeks to make with the monotheistic tradition in the name of the "pagan state" itself an expression of transcendent violence? Does not this extreme secularism, in declaring the biblical God to be evil, in corrupting all his traditions-philosophy, Christianity, Judaism, and why not Islam and in wanting to liberate the world from all of them, to liberate the world from history, perform a radical theological movement of transcendence, a Marcionite gesture, turning to a good world beyond the evil world, to a

hidden, loving God, completely separated from the revealed, wrathful God? And is not this combination of radical secularism and transcendence at the root of modernity and the modern state, as a combination of Lutheranism and secularism-the religion of the spiritual God separate from the human sovereign?

The important point for us is that Ophir's critique of the state leads us to the same conclusion as religious Zionism, namely, that Judaism is fulfilled in statehood. The Jewish state, in its present catastrophic violence, in brimstone and fire, is the figure of Judaism; it is the revelation of its God. This contention joins that of Ophir's previous book, co-authored with Ishay Rosen-Zvi, *Goy: Israel's Others and the Birth of the Gentile* (Ophir and Rosen-Zvi 2018), which found in rabbinic tradition the basis for racism in the State of Israel. In the new book, Ophir's secular response to the disaster of the Jewish state is the removal of Judaism from the state, that is exile.

## 2. Exile from Judaism

Exile appears in the title of Nottingham University and Polish Academy of Sciences philosopher Agata Bielik-Robson's latest book, *Derrida's Marrano Passover: Exile, Survival, Betrayal, and the Metaphysics of Non-Identity* (Bielik-Robson 2022a; see also Bielik-Robson 2014, 2022b). This book, too, does not deal directly with the Jewish state, but with the Judaism of the philosopher Jacques Derrida. However, in sophisticated Derridean readings of a variety of Derrida's texts, Bielik-Robson develops an independent and personal thought on contemporary Jewish identity. She continues Adi Ophir's secular response to the crisis of the Jewish state from the sources of Judaism.

The Jewish tradition to which the Warsaw philosopher turns is not the Bible, but the tradition of the conversos. The conversos were Jews who were forced to convert their religion by non-Jewish authorities and continued to live their Judaism in secret. The best-known example is the converts of the Catholic Inquisition on the eve of modernity in the Iberian Peninsula, who were also called New Christians or and this is the term Bielik-Robson chooses, following Derrida Marranos.

The question of the Marranos and their legacy has preoccupied contemporary Jewish historians and thinkers. In recent years, there has been a growing awareness of the unique customs developed by Marrano communities, some of which survive to this day, for example on the island of Mallorca. Benzion Netanyahu, the father of Israeli Prime Minister Benjamin Netanyahu, argued that the Marranos were mostly devout Christians and that their persecution as Jews by the Inquisition was carried out for ethnic reasons and heralded modern anti-Semitism, as also claimed by Yosef Haim Yerushalmi. Yitzhak Baer, on the other hand, saw the Marranos as Jews with a split identity, and Yirmiyahu Yovel discussed the Marrano split as a model for the consciousness of the modern secular subject.

Bielik-Robson follows Yovel in presenting the Marrano consciousness of hidden Judaism, alienated from itself, non-Jewish Judaism, not as a forced consciousness but as the core of a liberated, modern selfhood. Her book seeks "to defend the diasporic Marrano culture as a separate Jewish non-identity." (Bielik-Robson 2022a, p. 20) She does this by exposing the ideological foundations of the Marrano experience in the writings of Jacques Derrida, whom Bielik-Robson consecrates as the prophet of Marranism, "our new Marrano Moses." (id.) She herself identifies as a "*marrane* of Polish Catholic culture." (id.) By interpreting the term Marrano in a pun with the French word marre, "fed up," she strikes a provocative, protesting tone: "I grew very tired marre of constantly feeling apologetic about my own Jewishness." (Bielik-Robson 2022a, p. 19)

Note that Bielik-Robson's protest is not against Catholicism, which forced her to deny her Judaism, but against Judaism, which sees this denial as a problem. The Marrano Judaism that Bielik-Robson seeks to defend is based on the renunciation of Judaism, on the identity of non-identification. The paradox is expressed, among other things, in the designation of the negation of Judaism, from which Marranism lives, with the Jewish category of exile. The Marrano is exiled from her Judaism. Since for the Polish Catholic

Marrano this exile is not forced but chosen, she provocatively describes Marranism as an ethos of betrayal.

Bielik-Robson's Marrano betrayal of Jewish tradition has two faces that parallel the two sides of Adi Ophir's heretical theology: extreme immanent secularism combined with a Gnostic vector of extreme transcendence.

The first aspect of the Marrano exile from Judaism is secular. Like Ophir, Bielik-Robson opposes the metaphysics of religion, the world of values directed towards God outside the world. She too criticizes "the sacrificial logic inherent in all cultic religions," (Bielik-Robson 2022a, p. 52) the sacrifice of life for God. Betrayal is done in the interest of survival. "Instead of choosing *kiddush ha-shem* [martyrdom], a glorious death which sanctifies the Name, [the Marrnos] chose life as living-on, the sheer *survival*." (Bielik-Robson 2022a, 64).

The secular interpretation of the idea of exile as a distancing from Judaism was also offered by Jessica Dubow, a scholar of Jewish thought at the University of Sheffield. In her book *In Exile: Geography, Philosophy and Judaic Thought* (Dubow 2021), she points to a tradition of "exilic" thinking among 20th-century European Jewish thinkers. She understands exilic thought as critical thinking, which she finds, for example, in the way the prohibition of images served Walter Benjamin's anti-consumerist conception of the modern city, or in the way the idea of exile underpinned Franz Rosenzweig's critique of the nation-state and Hannah Arendt's critique of national, ethnic, or religious identity.

The secular thrust appears in the chapter on the British-Jewish intellectual Isaiah Berlin. Here Dubow formulates a critical exilic thought on the idea of exile itself. She employs Freud's distinction between melancholy and mourning as two opposing ways of living with loss, which she interprets as two opposing ways of existing in exile. The melancholic refuses to accept loss, reality. Dubow interprets the melancholic denial of loss as a reactionary exilic stance that rejects existing reality, rejects life in this world, and clings to utopian ideals. The secular critique of the transcendent God and sacrifice in Ophir and Bielik-Robson generates here anti-idealism.

Against the reactionary, melancholic exile, Dubow advocates an exile based on mourning. According to Freud, mourning does not refuse to accept loss like melancholia, but comes to terms with it and learns to live in the new reality. This is an exile that does not reject the real in the name of the ideal, but, on the contrary, abandons utopia in favor of pragmatism, as Isaiah Berlin did. Accordingly, Dubow's secular interpretation of the Jewish tradition of exile translates it as a project of Jewish assimilation into liberal society, the integration of Jews into the European state.

The category of exile serves Dubow to remove the idealistic, messianic sting from critical thought. One must ask what this analysis leaves of the concept of critique, and whether all that remains is not just the critique of realism against idealism, what is against what should be, that is, a critique of critique. A similar paradox appears here in the concept of Jewish exile, since Dubow proposes an exile whose essence is the abandonment of the exilic stance, the alienated, non-integrated stance that is, an exile from exile, a negation of exile.

The paradox of the secular Jewish position, which promotes not only non-Jewish Jews but also non-Jewish Judaism, brings us back to the Marrano theology of Bielik-Robson and reveals its second vector, not the immanent secular, but the transcendent spiritual. The exile from Judaism, the betrayal, is justified by the Derridean philosopher not only in the name of the value of life, which overrides the value of religion. Her more complex claim is that the abandonment of Judaism constitutes the essence of Judaism.

Bielik-Robson learns here from Derrida, who declared himself "the last Jew," the worst Jew, the last in his quality, the least Jewish, who therefore constitutes precisely the most authentic Jew, the last one left.

Bielik-Robson, like Dubow, does not build on pragmatism, but on messianic logic. The Marrano betrayal reveals itself as the true fidelity to Judaism by virtue of the "messianic reversal." The reversal is that the Messiah, the subject in whom the ideal of Judaism is embodied, is the one rejected by Jewish tradition as a heretic.

Bielik-Robson outlines a long Jewish tradition of heretical messianism. She quotes, "The stone the builders rejected has become the cornerstone" (Psalm 118:22), which was interpreted in the New Testament as fulfilled in Jesus. At the center she places the Lurian myth of God's exile, which she posits, following Gershom Scholem, as the seed of the sinning Sabbatean Messiah, the Polish antinomian legacy of Jacob Frank, and Derrida's Marrano Messiah. Scholem leads her further from Sabbateanism to Haskalah and secularism, which Bielik-Robson interprets as Marranism to the second power, Marranism from Marranism, exile from exile, which betrays Judaism even as a tradition of betrayal.

In Derrida, Bielik-Robson reads the final stage of Marrano eschatology: universalization. If Scholem pointed to the traitor Messiah as the secret of the modern Jew, she suggests that Marranism is the secret of the modern human. In her view, unlike conventional universalism, which promotes a unified positive concept of the human, Marrano universalism is based on a negative act of continuous self-denial, a selfhood based on "non-identification". The modern subject is a universal Marrano exile.

Within secularism and assimilation, at the heart of the sanctity of life, Bielik-Robson therefore preserves the negating exile as the secret at the heart of the Marrano subject, as the "non-" at the core of its identity. This secret crypto-Judaism constantly prevents full identification and perpetuates an inner alienation from the world. This is transcendence at the heart of immanence, reminiscent of inner faith as the religion of the secularized world, the religion of the hidden God, and it joins Marcion and Luther in Adi Ophir's pagan state.

One should ask whether Bielik-Robson's Marranism, insofar as it seeks to distinguish itself from "traditional" universalism, does not return to the mother of all Western universalist traditions, Pauline theology: "There is neither Jew nor Gentile, there is neither slave nor free, nor male and female; for you are all one in Christ Jesus" (Galatians 3:28), where the Messiah, the redeeming subject, is nothing but the infinite function of denying all identities.

The continuation of this thought continues to disturb: to understand the Marranos not as forced converts, but as voluntary ones, and their apostasy as the messianic realization of their Judaism, is this not precisely the ideology of the Inquisition and the Christian mission to the Jews? To Elliot Wolfson's criticism that, according to Bielik-Robson, Jesus is the last Jew, she replies, "Why not?" since for her heresy is the heart of Judaism. But she does not explain why her doctrine should not be seen as postmodern Christian theology and herself as a Lutheran Marrano in Jewish thought.

Some conversos became inquisitors, and betrayal also involves violence. This is revealed in Bielik-Robson's theology when she gestures at, quoting Derrida, "the 'death of Judaism, but also its one chance of survival'" (Bielik-Robson 2022a, p. 23). "Similar to Sabbatai Tsevi, who claimed that the ultimate goal of studying the Torah is the viola-tion of the Torah, Derrida will maintain that the final destiny of the Jewish archive is to turn into a heap of ashes," in which nevertheless "the embers keep glowing", but this is the glow of infinite extinction. (Bielik-Robson 2022a, p. 46) If Adi Ophir's book gestures toward the exile of Judaism from the secular state in response to the catastrophe of the Jewish state, the arguments of Agata Bielik-Robson and Jessica Dubow suggest that citizenship in the secular state is an exile from Judaism.

### 3. Everywhere at Home

Ophir, Bielik-Robson and Dubow generate opposition to the Jewish state by separating Judaism from secularism. A profound critique of this move from within the opponents of the Jewish state is formulated in Daniel Boyarin's latest book, *The No-State Solution: A Jewish Manifesto* (Boyarin 2023). A professor emeritus of Talmudic cultures at UC Berkeley, Boyarin is one of the foremost scholars of Judaism and one of the best-known Jewish intellectuals. In decades of prolific work, much of it in dialogue with his brother Jonathan Boyarin, an anthropologist of contemporary Judaism at Cornell, and in thoughtful engagement with cultural theory, Daniel Boyarin has developed a historical and conceptual vision of Jewish

existence. His new book offers a systematic articulation of his conception a "manifesto" as a summation of his work.

Boyarin's manifesto is explicitly directed against the Jewish state. His conception of Judaism is formulated as a response to the political hardship posed by the present existence of the State of Israel, first and foremost vis-à-vis the Palestinians. Defiantly rejecting the two conventional formulas for resolving the Jewish-Palestinian conflict, the two-state solution, one Jewish and one Palestinian, and the one-state solution of a single Jewish-Palestinian state, Boyarin proposes a "no-state solution", meaning no Jewish state.

Boyarin's rejection of the Jewish state is not, as in Ophir, based on a critique of Jewish tradition, but on the contrary, on that very tradition as Boyarin understands it. He begins by explaining what he does not mean. The opening move of his book rejects the conventional way of conceiving Judaism as opposed to the Jewish state or the idea of the state as such, that is, the notion of Judaism as a religion. Boyarin is taking aim at the dominant concept of religion, which refers to belief in a transcendent, otherworldly, metaphysical God. This concept of religion is based on an individual, inward faith, so that religion is seen as a private matter between a person and God, outside of any public or political dimension, outside of any cultural or historical context.

Boyarin's opposition to understanding Judaism as a religion is based on his earlier work, in which he contributed to the contemporary critique of the general use of the category of religion. At the heart of the critique is the claim that the concept of religion in the sense of faith describes a specific cultural formation, European Protestant Christianity, based on a Pauline conception of pure inward spirituality. Abstracting religion from its European context, positing it as a general concept, and applying it to other contexts, so the critical claim, constitutes cultural imperialism.

This was the claim made by Tomoko Masuzawa in her 2005 book *The Invention of World Religions, Or: How European Universalism Was Preserved in the Language of Pluralism* (Masuzawa 2005) with respect to non-European cultures, and by Boyarin herself, along with Carlin Barton, in their 2016 book *Imagine No Religion: How Modern Abstractions Hide Ancient Realities* (Barton and Boyarin 2016), regarding pre-modern cultures. In his latest book, Boyarin shows how European cultural imperialism, which seeks to reduce world cultures to private beliefs through the propagation of "religion," complements its conquest of the public sphere with a discourse of cosmopolitan universalism that erases all cultural difference.

Boyarin's opposition to Pauline religion goes even deeper, touching on the historical tension between Judaism and Christianity. Boyarin has shown that religion as an inward belief developed in Pauline Christianity through the redemption of the pure spirit from the impure flesh, which has been historically interpreted as the liberation of Christianity qua inner Judaism from outer Judaism law, culture, nation, and state. The cultural imperialism of Christian Europe emerged in late antiquity in the form of supersessionist theology, according to which spiritual Israel, Christianity, replaced Israel in the flesh.

Boyarin fights against the erasure of Judaism through the allegorization of outer Jew by inner Jew, flesh by spirit, in a long tradition from Paul's own writings to modern French thought, in Nancy or Lyotard, for example (Boyarin 1994). To this list one can now add the Derridean Marranism of Bielik-Robson, which preaches the renunciation of Judaism as tradition in favor of Judaism as secret in the heart of the subject, as well as Dubow's Jewish exile as exile from Judaism.

When Boyarin calls for a Judaism without a state, he does not mean Judaism as a pure inner spirituality, as a religion. On the contrary, his book is a radical manifesto for Judaism as a nation. He rejects the notion of the Jewish nation as based on "Judaism", a system of ideas that he identifies as the basis of Paulinism. Boyarin does not speak of Judaism, but of Jewishness, as the national culture of the Jews, what he calls Yiddishkeit, which is the antithesis of religion. The Jewish nation is not a community of believers, a church, or a school of thinkers, but a family, a group of people united not by an idea but by blood ties. "Existence without prior essence: a Jew is a Jew is a Jew, not one who believes this or does

that, but simply one who is born to a certain people anywhere or has become naturalized into that people" (Boyarin 2023, p. 51).

If Paul's religion stands for purity of spirit, Boyarin's Jewish nation stands for purity of flesh. His Judaism is devoid of any transcendence, of any God or faith, and in this sense it is secular. If Ophir and Bielik-Robson oppose Judaism to secular existence, Boyarin posits Judaism itself as secularism.[1]

Boyarin's anti-Pauline move from spirit to flesh is reminiscent of Nietzsche's anti-Platonic move from the metaphysical to the physical, from the idea to life. One may wonder here, as Heidegger did with respect to Nietzsche, whether the inversion of Paulinism is not after all just an inverted Paulinism, one that preserves the distinction between spirit and flesh, between religious and secular. Talal Asad has already noted that religion, in the sense that Boyarin criticizes it, is itself a secular category.

To be sure, Boyarin's Yiddishkeit is not just flesh, devoid of any ideal content, any Judaism. "Existence without prior essence" is not devoid of any essence, only a prior one, but it has an essence that follows its existence. Judaism is the culture developed by the Jewish nation. To explain his conception, Boyarin turns to Jean-Paul Sartre's existentialism, according to which human consciousness arises from the specific situation into which each person is "thrown".

Sartre described the emergence of Jewish consciousness from the Jew's situation of anti-Semitism, and of black consciousness from the black person's exposure to racism. Sartre's thought inspired black thinkers—Aimé Césaire, Léopold Senghor, Frantz Fanon— to develop the idea of black culture, *négritude*. Boyarin applies the same idea (a spiritual world arising from a given existence) to Jewish culture. Yiddishkeit is *judaïtude*, which Boyarin, inspired by Judith Butler's gender theory, defines as "the performance of being Jewish." Unlike Sartre, who limited Judaism to anti-anti-Semitism, Boyarin places the Jewish soundscape at the center of Jewish performance: Yiddish and Jewish multilingualism in general, along with the discourse of the Talmud, "the music of Jewish life," which Boyarin describes as jazz.

It is doubtful that Boyarin finds support where he looks. The main thrust of the thought of Sartre, Fanon, and Butler is not the subjugation of the spirit to the flesh, but just the opposite. They show that the supposedly natural, biological fact-skin color, gender-acquires meaning and existence only as a social construction. Race and gender are not biological givens, but political-conceptual positions.

Is Jewishness really an existence without a prior essence? Is there such a thing? Even if not a Pauline inward spirituality, has not Jewish historical existence-including the familial-depended on some narrative, value, or idea, in short, on Judaism? Does not the Jewish nation, like the black race, embody a political positioning? If not, is this nation not somehow still a kind of religion, a private family matter without political significance that does not interfere with cosmopolitan universalism?

And yet the title of Boyarin's manifesto is political: "No State." The Jews are a nation without a state; they are in *galut*. Does Boyarin breathe into carnal Israel the spiritual denial of the state? The answer is no, and the root of the problem lies in understanding the meaning of "no". Boyarin insists that the rejection of the Jewish state does not mean that Jews lack statehood. He rejects the understanding of the concept of Jewish *galut* in terms of exile, that is, the lack or absence of a state or home. According to him, understanding the Jewish nation as lacking a state means that Jews miss the state, that is, they want a state. As exile, *galut* is not the opposite of a Jewish state, but rather the longing for one.

Hence Boyarin's conception, developed in the past, that Jewish *galut* should not be understood negatively, as exile, but positively, as *diaspora* (Boyarin 2015). Unlike exilic, stateless existence, which is at home nowhere, Jewish diasporic existence is, on the contrary, at home everywhere. Moreover, it has a double home, in two spaces. First, the Jew feels at home in Jewishness, in the diasporic space of the Yiddishkeit family. Second, the Jews feel at home in their place in the local space, each person in their own "here," hence the

principle of "hereness," *doykayt* in Yiddish, which Boyarin borrows from the language of the Bund.

But the Bund—the General Union of Jewish Workers in Lithuania, Poland, and Russia was defined not only by Jewish flesh but also by the spiritual, socialist workers' ethos. The socialist struggle determined whether the Jewish worker was "here," at home, or not. Since Jewishness is defined in Boyarin's conception as "existence without prior essence," it is not clear what idea or ideal will determine the Jews' identification with their place. In their diaspora, the Jews are always at home, in every "here," whatever its essence may be. In other words, the Jewishness of the UC Berkeley Talmudist does not in itself seem to provide any political principle, negative or positive.[2].

Boyarin's Diaspora Jews, with no Jewish state, have all the other states—American, German, French. In this sense, Boyarin agrees with Ophir, and also with Bielik-Robson and Dubow, that Jews are at home in the modern state. Unlike Ophir, who criticizes the nature of the sovereign state, Boyarin's stance is not against the state as such, but against the mono-national state, accepting any multicultural political framework, including empire and multinational state. Boyarin's Jewish nation feels so much at home in the state that it is unclear why it should not also have one in the "here" of the Land of Israel, if not as a Netanyahu-style Jewish state, then as a Herzlian state, a (Boyarin quotes Dimitry Shumsky) "non-Jewish state for Jews".

And maybe there already is one?

## 4. State of Exile

This is what Danny Trom, a sociologist at the School for Advanced Studies in the Social Sciences in Paris (EHESS), claims in his new book *State of Exile: Israel, the Jews, Europe* (Trom 2023). In Trom's book, the idea of Jewish exile is interpreted not as resisting the State of Israel, but as justifying it.

The way in which Trom characterizes Jewish existence is close to that of Boyarin, and also to that of Bielik-Robeson and Dubow, that is, the secular conception. For Trom, Jewish existence is not defined by an idea, an ideal, or a metaphysical affiliation; Jewishness is not in the spirit but in the flesh. Boyarin spoke of "existence without prior essence"; Trom speaks of "the Jewish fact," *le fait juif*.

Trom's secularism is extreme. If, for Boyarin, Jewish existence nevertheless developed the essence, culture, and spirit of Yiddishkeit, and the non-Jewish Jew of Bielik-Robson and Dubow retains inner transcendence, Trom's Jews exist simply as a fact of nature, as an organism. Their only concern is to be, that is, to survive.

The only content of Jewishness that Trom discusses, the only question that preoccupies it and him, is the struggle for survival against the threat to the existence of the Jewish fact. For Trom, Judaism is entirely defined by anti-Semitism. This is a Jewishness whose entire soul is resistance to its negation, Jewishness as anti-anti-Semitism, an attitude that was greatly strengthened among Jews after October 7.

The political reality that emerges in Trom's book, as developed in earlier books (Trom 2019), is therefore Schmittian. For Trom, as for Carl Schmitt, the political collective is defined by the life-and-death struggle with its enemies. This leads to the inevitability of war, which Schmitt based on a political theology of original sin, according to which man is evil by nature until he is redeemed from sin by the second coming of Jesus. The pre-redemptive human existence, in which evil is necessary and war is natural, is marked in Trom's language by the concept of exile.

For Trom, as for Schmitt, politics enables survival in an evil world, that is, it offers protection. The polity is a mighty force against evil, what Hobbes called "Leviathan" and Schmitt identified with the sovereign. Trom draws on Yosef Hayim Yerushalmi, who described historical Jewish politics as a "vertical alliance" of the Jews with the sovereign, the prince or king, against the Jews' enemies. It follows that the Jews are the paradigmatic citizens of the Schmittian state, which strengthens Adi Ophir's thesis that the modern state is the true contemporary manifestation of biblical theo-politics.

Trom adopts Jean-Claude Milner's claim that the great crisis in modern Jewish history was the rise of the democratic republic, in which the people became sovereign. The disappearance of the sovereign as one elevated above the people, as a king, eliminated for the Jews the possibility of defending themselves against their haters through a vertical alliance. Democratic Europe exposed the Jews to genocide. The only solution was to "divorce Europe," that is, to establish a separate state for the Jews, and that was the purpose of Zionism.

Trom dismisses the socialist-Zionist visions of creating a better society, of tikkun olam, in the spirit of Buber, as a childish "petite bourgeoisie." The Parisian sociologist focuses on Herzl's political Zionism. Like Boyarin, Trom emphasizes that Herzl did not speak of a nation-state or a Jewish state. But while Boyarin reads Herzl's Jewish state as a diasporic autonomy in a multicultural space, Trom reads it as a refuge against anti-Semitism. Zionism did not seek to create a Jewish state that would redeem the Jews from exile; on the contrary, it sought to create a Leviathan, a sovereign in the European style, that would protect the Jews in their continued exile. The Zionist creation, the State of Israel, is not the negation of exile, but the "state of exile".

Trom's position contradicts the conception of Israel as the nation-state of the Jewish people and offers a secular opposition to current Israeli chauvinism, as in the Basic Law Nation-State and the intensifying religious nationalism. A major difficulty is that Trom's conception, even if it is faithful to Herzl, does not fit the reality of the State of Israel, at least since its establishment.

Particularly perplexing is his assertion that the State of Israel, since its role is to protect Jews in exile, should be understood not as a project of national self-definition but as part of the international effort to protect minorities and refugees. This assertion ignores the enormous enterprise of creating Israeli national existence. To ignore Israeli nationhood is to ignore the reality of the State of Israel. Worse, this claim ignores that the establishment of Israel created the Palestinian refugee crisis, which has only worsened over the years, and today more than ever, in defiance of all international protection mechanisms.

Justifying the persecution of the minority in the name of protecting the majority in the state as a minority in the world is an abomination not only in Zionism, but in the nation-state in general. In fact, Trom's "state of exile" is not at all opposed to the nation-state, but reflects its logic, that is, to protect the national group threatened by imperial power or other nations. The self-conception of the sovereign majority as a persecuted minority—"exile"— produces endless violence in the name of endless self-defense, creating more and more persecuted, refugees and exiles. The mystifying discourse of liberal Zionism is worse than the open racist messianism of religious nationalism, which at least calls reality by its name.

***

The positions presented so far are opposed to the religious-national vision of the Jewish state on two pillars. First, they are secular in the sense that they are committed to a worldly existence without reference to divinity, that is, to metaphysical otherness. Ophir, Bielik-Robson, and Dubow contrast secularism with Judaism, while Boyarin and Trom reconcile them. Second, for these authors, Jews are at home in the modern state. This homeliness is paradoxically based on the concept of exile—exile from exile in Bielik-Robson and Dubow, exile as diaspora in Boyarin, and exile as sovereignty in Trom.

An agreement between the religious supporters of the Jewish state and its secular critics emerges here. A number of thinkers have addressed this agreement and the difficulties it poses. Menachem Lorberbaum, professor of philosophy at Tel Aviv University and one of the leading Jewish theologians of our time, co-editor with Michael Walzer of the series "The Jewish Political Tradition," recently pointed out that not only in Israel, but also in America, Jews "feel at home" and do not experience exile in the sense of not belonging to the state. Shaul Magid, a professor of Jewish studies at Dartmouth College and an expert on Hasidism and contemporary Judaism, argues in a new book that the abandonment of

exile has robbed liberal Zionism and Diasporism of the ability to effectively counter the chauvinistic patriotism of religious Zionism.

Accordingly, in their recent writings, Lorberbaum and Magid, along with Amnon Raz-Krakotzkin, have developed another front of Jewish opposition to the Jewish state. In contrast to the movement that uses the concept of exile to domesticate Judaism in the secular state, these thinkers emphasize in the idea of Jewish exile a non-secular dimension in which they anchor a radical critique of the modern state.

## 5. Phenomenology of Exile

Menachem Lorberbaum developed his thoughts on the subject in an important article, "A Theological Critique of the Political" (Lorberbaum 2023; see also Lorberbaum 2020). Like Adi Ophir, Lorberbaum formulated a political-theological critique of the modern state that focused on the figure of the sovereign. The Jewish God also plays a key role in Lorberbaum's analysis. But unlike Ophir, for whom "the state is monotheism's true contemporary figure," that is, the modern sovereign is the manifestation of the transcendent God, Lorberbaum agrees with Schmitt that secularization separates the sovereign from God: the sovereign is not God but his substitute.

According to Lorberbaum's analysis, the institution of the modern state usurps the absolute status of the extra-worldly God; it usurps the sovereignty of the world's sovereign and vests it in a worldly authority, man. The sovereign of the modern state, the Leviathan, is, as Hobbes said, a "Mortall God." Unlike the Catholic Schmitt, who saw the secular sovereign as a necessary substitute for God in His absence, Lorberbaum sees the secular state as a declaration of the "death of God." The sovereign does not take God's place, but steals it.

The transition from divine to state sovereignty, which for Adi Ophir is identity and for Carl Schmitt is translation, for Menachem Lorberbaum constitutes idolatry. This is his theological critique of modern sovereignty. "Sanctifying a secular institution" is the idolatry that Lorberbaum also attributes to religious Zionism, which, in the spirit of Rav Kook, has dedicated itself to the cult of the Jewish state. The problem, he emphasizes, is not only theological; it is not only an apostasy from God. The deification of the state, Lorberbaum agrees with Ophir, sanctifies the absolute power of the sovereign and produces unlimited violence, which in the Jewish state is manifested in the endless oppression of the Palestinian people.

Whereas Ophir's position locates this violence at the foundation of Judaism ("In the beginning was the state"), as opposed to secularity, Lorberbaum identifies the source of violence in secularization and finds in the sources of Judaism not the state but resistance to it. Lorberbaum's source is not the Bible, but the post-biblical rabbinic tradition built on the ethos of exile.

To understand the idea of exile as a critique of the modern state, Lorberbaum turns to modern Jewish exilic thought. He refers to Kabbalah, which in its modern Lurianic manifestation, as we have seen in Bielik-Robson, attributes exile to God Himself, expressing His concealment, His transcendent otherness. The theological exile is translated by Lorberbaum on the human level into a "consciousness of exile" that produces a "political sagacity" in rabbinic Judaism.

Rabbinic exilic political wisdom, as Bielik-Robson and Danny Trom suggest, aims at the survival of the Jews in the Diaspora. However, in contrast to Trom's Schmittian logic, for Lorberbaum the survival of the Jews in exile does not mean the preservation of their organic existence through an alliance with sovereign violence, but the preservation of their exilic existence, that is, an existence that does not submit to the logic of sovereign power. Unlike Boyarin, who interprets exile as the absence of a state whose purpose is a state, Lorberbaum interprets exilic statelessness as a resistance to the state that survives within the state.

In contrast to the secular vision of Boyarin and Trom, Lorberbaum's rabbinic exile project draws on God's transcendent otherness. Divine metaphysics is not translated here,

as in Ophir and Schmitt, into sovereign absolutism, into lawless power. On the contrary, for Lorberbaum, transcendence is an anchor for an ethical ideal beyond political reality. Here he follows Jewish thinkers such as Hermann Cohen and Emmanuel Levinas, for whom the relationship to the extra-worldly God provides a "moral compass" outside the power field of the state. Beyond a carnal existence concerned with self-preservation, the relationship to otherness offers Judaism a different kind of existence, an ethical one.

Bielik-Robson narrows the relation to transcendent otherness to an inner secret in the individual that prevents any collective identification, that cuts off all belonging. In contrast, Lorberbaum (again, like Cohen and Levinas) measures the power of exilic existence precisely in its ability to create community. He draws from the Jewish political tradition the institution of the kahal, a community based on brotherhood and mitzvot, as opposed to the state body held by sovereign power.

Lorberbaum sees the exilic kahal as a model for a multicultural civil society that resonates with Boyarin's diaspora and sustains a moral reality that limits state violence. He also suggests an alternative direction for the Jewish state project that is non-secular but opposed to nationalist religiosity whose vision is territorial sovereignty. Like Levinas and Buber, Lorberbaum argues that understanding the Land of Israel as holy land nullifies any claim to sovereignty and entrusts the government of the land not with ownership but with stewardship, that is, the duty to ensure justice, not only for Jews but for all citizens, and not only for them but also for the stateless: the indigenous, the uprooted, the refugee, and the immigrant.

## 6. The Necessity of Exile

Lorberbaum outlined the theo-political contours of opposition to the Jewish state based on the Jewish exile tradition. The application of this idea to the current situation—until October 7—of the Jewish discussion on the State of Israel is provided by Shaul Magid in the essay collection *The Necessity of Exile: Essays from a Distance* (Magid 2023).

Magid joins Lorberbaum and Boyarin in diagnosing the acute problem of the Jewish state project as the injustice to the Palestinian people-oppression, occupation, dispossession-in the name of the sovereign claim to exclusive Jewish ownership of the land. His alternative vision is one of Jewish-Palestinian coexistence. According to Magid, both the settlement movement and the BDS movement, the former in Jewish discourse and the latter in Palestinian discourse, have erased the Green Line, the separation between Israel and the occupied territories, thus sealing the fate of the two-state solution. In reality and in imagination, Jews and Palestinians live under one rule. The choice today is between apartheid rule or liberal democracy, a state of all its citizens, in the spirit of Brit Shalom.

Magid speaks of one state, liberal, and sounds less extreme than Boyarin, who says "no state". However, his opposition to the Jewish state leads him to an even more extreme contrast between Judaism and the state. He sees the need for *galut* not as diaspora but as exile, that is, as Judaism essentially opposed to the state. More precisely, the need is to renew exile, that is, to negate its negation. Magid, like Lorberbaum, identifies the negation of exile in both liberal Zionism and American diasporism, which do not resist religious Zionism enough because for them, too, Jews are at home in the state.

Magid's essays trace the idea of exile in modern Jewish thought in a variety of forms. In Peter Beinart and Judith Butler, he identifies a "new secular Jewish identity" (Magid 2023, p. 65) that is premised on a diasporic existence. From Bashevis Singer he draws a spiritual Judaism without territorial sovereignty, and from Martin Buber a possession of the land that is not sovereign, not ownership, but (as Lorberbaum says) rather stewardship.

Particularly important is a quote from Reform rabbi Eugene Borowitz, who published *A Theology for the Post-Modern Jew* in 1991: "Anybody who cares seriously about being a Jew is in Exile and would be in Exile even if that person were in Jerusalem. That Exile results because our Jewish ideal is unrealized anywhere in the world." (quoted in Magid 2023, p. 3) This is the idealism that Lorberbaum found in Levinas and that Jessica Dubow rejected in

the name of Isaiah Berlin's pragmatism: exile as clinging to an ideal beyond reality, what can be called a pre-redemptive messianic existence.

Here the connection between the relation to a transcendent otherness beyond the world, beyond secularism, and the consciousness of exile as resistance to the given order is sharpened. The modern Jewish thought in which Magid finds the deepest consciousness of exile is therefore that which has offered the most significant resistance to the modern world order, namely Haredi Judaism. Magid's discourse, as well as that of Lorberbaum and Raz-Krakotzkin, joins a post-secular trend that sees secularism not as rebellion but as hegemony, and therefore looks for critical thought not on the secular side but on the side of tradition, Orthodoxy, or ultra-Orthodoxy (See for instance, Milbank et al. 1999).

*The Necessity of Exile* proposes to rethink the importance of Haredi culture as a resistance to the false redemption offered by modernity in the sovereign state. According to Magid, the Haredi rejection of secular-liberal assimilation in the style of Bielik-Robson and Dubow is not an apolitical rejection of the world, as it is often portrayed. The thinkers he discusses show a deep political awareness of resistance to state violence, similar to that formulated by Adi Ophir and Menachem Lorberbaum. This is the basis of Haredi anti-Zionism, which rejects the state and its military as the quintessence of secular violence.

Two prominent figures stand out in the book. The first is the founder of Satmar, one of the largest Hasidic groups, Rabbi Yoel Teitelbaum, a well-known anti-Zionist Haredi. Teitelbaum laid out his doctrine in several writings, including *Vayoel Moshe* (1961) and *Al Hageulah ve-Hatmurah* (1967), a selection of whose ideas and translations will appear in Shaul Magid's forthcoming book. The Satmar Rebbe criticized Zionism for its worship of sovereign national power as false messianism and heresy. In contrast, he posited the necessity of exile as a messianic act of spreading Torah throughout the world in preparation for redemption, a project also promoted by Menachem Mendel Schneerson, the Lubavitcher Rebbe, leader of the Chabad Hasidic movement.

The second, lesser-known figure is the Lithuanian Rabbi Aharon Shmuel Tamares. He rejected Zionism "not because it is secular or even a form of premature messianism, but because it is violent." (Magid 2023, p. 217) Tamares is an example of a critique of the modern state from traditional Jewish exile thought. He was, Magid writes, "more focused on what Judaism could offer Europe in its time of nationalist crisis, that would soon explode into a world war, rather than what European political philosophy had to offer a fledgling Jewish state." (Magid 2023, p. 221) Judaism offers exile as a "collective positionality outside the realm of nation-states," (Magid 2023, p. 216) that is, social responsibility beyond state interests. Are anti-Zionist Haredim the embodiment of the nonviolent ethical community envisioned by Levinas and Lorberbaum?

Magid, however, is not anti-Zionist, but "counter-Zionist," meaning that he does not use exilic thinking to oppose the State of Israel outright, but rather to correct it. The key thinker he relies on is Shimon Gershon Rosenberg, Rav Shagar, whose essay "On Religious Post-Zionism: A Sermon for Independence Day," Magid translates in his book.

The essay is directed against religious Zionism in the spirit of Kook. Shagar criticizes Kookism as modernism that abandoned exile and positioned itself as redeemed, master of the world, as a force of nature. Like Boyarin, he condemns as oppressive the cultural unity that the modern state seeks to impose. Shagar called the alternative "post-Zionist religiosity" or "religious post-Zionism. It combines the mysticism of Rabbi Nachman of Breslov with postmodernism's embrace of difference and multicultural plurality. The idea is to undo Zionist hegemony and create equal space in the country for haredi and Arab cultures. Magid describes Shagar's vision as an "exile in the Land [of Israel]" whose purpose is Jewish-Palestinian coexistence.

One might ask how much exile is really necessary for Magid's vision of a binational state. Martin Buber and Brit Shalom did not advocate exile, but socialist redemption. Shagar, too, wrote of a "paradoxical peace that sees the other, the Arab, as belonging to the homeland, without relinquishing the feeling of being at home in one's homeland" (Rosenberg 2014, p. 176).

To the extent that the renewal of exile means alienation from the institutions of the liberal state, it cuts both ways. Magid locates the rise of post-Zionist religiosity in the rift that emerged between the state and religious Zionism in the wake of Israel's 2005 disengagement from Gaza. In this rift, settlers positioned themselves against the state like indigenous people defending their land against foreign forces. Magid's personal narrative of his journey as a young American in and out of the settlement movement reveals the settlements as a counterculture, "odd amalgam of Rav Kook, Aaron David Gordon, and Bob Marley." (Magid 2023, p. 41) The Contra-Zionism that emerges from this cocktail could foster citizenship without sovereignty, in the spirit of Menahem Forman, who promoted Jewish-Palestinian neighborliness. On the other hand, as Schmitt's prophecy is fulfilled before our eyes, the settler rebellion could take over the state, unleash sovereign power from all restraint, and in racist messianic intoxication perpetrate what Ophir describes as divine violence, that is, a holocaust.

## 7. Exile in the Land of Israel

The above review revealed two contrasting trends in current opposition to the Jewish state in the name of exile: one rejects the Jewish state as a religious idea in favor of secularism; the other rejects the Jewish state as a secularized religion in favor of Jewish tradition. Amnon Raz-Krakotzkin, professor of Jewish history at Ben-Gurion University, who in the 1990s wrote an essay that became a canon of thinking about exile (Raz-Krakotzkin 1993, 1994), in his new book *Mishna Consciousness, Bible Consciousness: Safed and Zionist Culture* (Raz-Krakotzkin 2022), offers a complex analysis for understanding the deeper connections between the two trends and how the first, secular, falls under the critique of the second, post-secular.

Raz-Krakotzkin's intervention is not only about history, but also about historiography. In contrast to linear historiography, he draws inspiration from Walter Benjamin to write history as struggle, "against the grain." In the shadow of the hegemonic narrative through which the rulers of the present find their future in the past, the Benjaminian historian searches the past for moments when the hegemonic continuum is disrupted. He uncovers events that are not causes for the present situation of power, but that resist power and thus constitute "a hidden indication towards redemption." (Raz-Krakotzkin 2022, p. 218). This is the theo-political essence of post-secular historiography.

The historiographical struggle that Raz-Krakotzkin wages revolves around the understanding of modernity itself. As the title of the book suggests, it is a struggle between two consciousnesses, two narratives, one hegemonic and one in its shadow. Both narratives are theological in that they give meaning to modernity as a time of God's absence, when God has disappeared from His place in the world, from the center of culture. The contention of the book is that the two modern consciousnesses are connected to two consciousnesses of Jewish existence in Israel/Palestine, constructed on two textual traditions: "mishnaic consciousness, biblical consciousness." The Bible is the foundation of a hegemonic modern consciousness that not only competes with the Mishna consciousness, but erases it. Raz-Krakotzkin's historiography resists this erasure, first of all by exposing it.

The hegemonic narrative belongs to the dominant modern Western culture. The dominant Western consciousness makes sense of the modern absence of God through a secular-Protestant amalgam. Raz-Krakotzkin refers to Max Weber. For us, this amalgam was revealed in the thought of Ophir, Bielik-Robinson, and Boyarin. It is a dualistic movement that, on the one hand, grasps God's absence as His liberation from the world into radical transcendence, accessible only through an individual inner spirituality: the Pauline faith. On the other hand, God's liberation from the world also liberates the world from God, which is interpreted as the secular redemption of matter.

Western modernity understands God's absence as the liberation of man to feel at home in the world and to do with it as he pleases. Hence capitalism for Weber, technology for Heidegger, and hence what Raz-Krakotzkin calls the millenarianism of progress that drives colonial expansion and sustains the absolutism of state sovereignty. Raz-Krakotzkin's cri-

tique of Western modernity places him at odds with secular projects such as Bielik-Robeson's Marranism, Dubow's assimilationism, Boyarin's diasporism, and Trom's Schmittianism.

Raz-Krakotzkin proposes that the theo-political narratives of modernity are expressed in conceptions of the Jewish connection to Israel/Palestine and to Judaism itself, the Jewish text. The Western secular-Protestant narrative sees modernity as man's salvation from religion, but also religion's salvation from human history, by returning to religion's source in the Bible, purified of all interpretive tradition, sola scriptura.

The biblical source from which modern national liberation finds its model is the conquest of the Promised Land by the Israelites. Hence the paradigmatic role in Western consciousness of the return of the Jewish nation as an exiled sovereign to its land, or Zionism. Lutheran secularized modernity is embodied in the consciousness of Jews who, in the mirror of the Bible, see themselves as the people of Joshua, son of Nun, and Palestine as the land of Canaan. The Zionist conquest re-enacts biblical geography as a model for European colonialism. This analysis confirms the link between the ethos of the Jewish state and the secular opposition to it of the model that I outlined through Ophir's book, which, like Luther, brings Jews back to their biblical source as the beginning of their redemption from Judaism.

Western modernity perpetuates its domination by hiding its alternative and resisting this concealment is at the heart of Raz-Krakotzkin's book. In the shadow of the Western narrative, he points to traces of another, repressed, Oriental modern existence. In contrast to the Protestant West, Oriental modernity is Muslim in character, and the book points to its Western affiliation with Catholicism, to which we could add Eastern, Orthodox Christianity.

This modern consciousness does not understand God's absence as a severing of transcendence from immanence, as the redemption of the world, but, on the contrary, as an intensification of the tension between the world and God, sharpening the awareness that the world remains unredeemed. Through this narrative, man does not feel at home in his world, as master of it, but rather as a stranger, in exile. He does not rejoice at the death of God and surrender to the body of the present, but clings all the more fiercely to any means that mediates him to the absent divinity. Oriental modernity is committed to tradition.

The Jewish embodiment of Oriental consciousness is exilic, and exile par excellence, modern, is Israel's exile from its exile (in Spain) in its land. The "hidden indication" that Raz-Krakotzkin's Benjaminian historiography redeems from Zionist historiography is a concentrated moment of Jewish exilic community in Ottoman Palestine in the third quarter of the sixteenth century, in the city of Safed. "The moment of Safed" gathers a group of figures who shaped the face of early modern Judaism after the expulsion from Spain, led by Rabbi Joseph Karo, author of Beit Yosef and Shulhan Arukh; Rabbi Isaac Luria Ashkenazi, father of Lurianic Kabbalah; Kabbalist Rabbi Moses Cordovero; poet Rabbi Israel Najara; Rabbi Solomon Alkabetz, and others.

Unlike Zionism, which is characterized by a Lutheran return to the biblical kingdom of Judah, the Safed consciousness relates Jewish existence in Israel after the destruction, whose place is not Jerusalem but the Galilee, and whose discourse is not prophetic but rabbinic. This is not Bible consciousness but Mishna consciousness, not source but tradition. More precisely, it is the tannaitic consciousness that in the Safed discourse founded the tradition not only as Mishnah, but also as Kabbalah, given in the Zohar by Rabbi Shimon Bar Yohai.

The Safed nation is not founded on Joshua son of Nun conquering the land, but on Rashbi maintaining the exile in the Land of Israel. The connection through Rashbi between Mishnah and Kabbalah, Karo and Luria, law and narrative, is central to Safed's vision of exile, according to Raz-Krakotzkin. It opposes the common narrative from Gershom Scholem's school of the Lurianic exile of the Shekhinah as the seed of Shabbetai Tzvi's heretical Messianism and from there to secular Judaism, a foundational narrative in both Zionism and Bielik-Robeson's Pauline Marranism à la Derrida. In Safed, Raz-Krakotzkin argues, the awareness of God's exile does not lead to His abandonment and the proclamation of secular redemption, but, on the contrary, to a sharper understanding of

the world's corruption and the need to repair it through mitzvot. Luria's core legacy is not Shabbateanism, but Shulhan Arukh.

Against the Jewish state, Lorberbaum posits the exilic community as a pluralistic civil society; for Shaul Magid, exilic Judaism means opposition to the state by anti-Zionist Haredim or counter-Zionist settlers. For Raz-Krakotzin, modern Jewish exile is embodied in Mizrahi traditionalism, which resists not only the Jewish state but also the political theology of the West. *Mishnaic Consciousness, Biblical Consciousness* locates Jewish opposition to the Jewish state in Ottoman Safed, that is, in the Muslim-Arab space that is Israel's concrete space today.

*** 

Political sovereignty represents the ultimate vision of the modern political imagination. The establishment of the state is perceived as the moment of liberation, independence, redemption from enslavement, and the end of exile. However, as Danny Trom has noted, the European sovereign state does not offer redemption, but rather shelter in a world without redemption, a world whose law is war.

The mistake that perpetuates evil is not only to confuse the state with redemption, but to see it as a refuge from war. Because the state itself, as Adi Ophir and Menahem Lorberbaum explain, is war; it is the Leviathan that does not protect from violence, but is its embodiment. And the harshest manifestation of the violence of the modern state was not the war between states, the world war, but the violence of the European state against what is perceived as outside it, as outside the state and the laws of war: such as the European Jews and non-European peoples.

In this sense, the state of Israel is a European state. Jewish sovereignty in the Land of Israel, supposedly redemption, supposedly refuge, has in practice been the negation of its otherness. First and foremost, this is the non-European, the Palestinian. As Raz-Krakotzkin shows, Zionism did not negate Europe; on the contrary, it carried out the European negation of Islam and the Arab world by erasing Palestine. The existence of Israel is the existence of the Nakba, the Palestinian catastrophe, which is not in the past but in the present, and its name today is Gaza.

But the state of Israel is also European in its negation of the Jews, from both of its locations, according to Boyarin's geography. First, the sovereign Jewish state alienates the Jews—exiles them—from their "here" in the Land of Israel, that is, from their concrete existence in the Middle East. The negation of Palestine perpetuates Israeli Jews as occupiers, settlers, and colonialists. Second, Jewish sovereignty negates Judaism itself as a tradition of exile, not in the secular sense of diaspora or lost sovereignty, nor in the religious sense of apolitical spirituality, but as resistance to state sovereignty and its violence.

The contemporary meaning of exilic thought is the negation of the three negations: of Palestine, of the Jews, and of Judaism. The idea of exile aims to end the negation of Palestine without the genocide of the Palestinian people by limiting Jewish sovereignty in the Land of Israel. The idea of exile aims at limiting Jewish sovereignty without expelling the Jews from the Land of Israel, by integrating the Jews into the Arab and Islamic space. The idea of exile aims at integrating the Jews into this space without erasing Judaism, but on the contrary, by restoring it as a tradition of social existence whose essence is not sovereign power.

**Funding:** This research received no external funding.

**Data Availability Statement:** No new data were created or analyzed in this study. Data sharing is not applicable to this article.

**Conflicts of Interest:** The author declares no conflict of interest.

**Notes**

[1]  Jewishness as the inversion of Paulinism was described in Boyarin's earlier books, most famously in *Carnal Israel: Reading Sex in Talmudic Culture* (Boyarin 1993).

[2]  Julie Cooper described this, comparing Boyarin's diasporism to Simon Dubnow's, as "reducing a powerful repository of political templates to a dissident subculture" (Cooper 2023).

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
