# Peer review of "Back to Exile: Current Jewish Critiques of the Jewish State"

_religions, doi:10.3390/rel15020250_

Round 1

Reviewer 1 Report

Comments and Suggestions for Authors

This is a highly sophisticated and well-informed essay on a timely and important topic.  I consider this original and pioneering article to be a particularly welcomed contribution to the current intellectual and scholarly discourses on contemporary Judaism and its relations with the State of Israel.  The essay is well written, well argued, and lucid. In spite of rich material and extensive analyses of philosophical-theological tracts, it makes for easy and enriching reading.  I embrace this article enthusiastically. 

Author Response

Thank you for this encouraging review.

Reviewer 2 Report

Comments and Suggestions for Authors

The author commences the discourse with considerations rooted in personal and emotional contexts, which are not aligned with the criteria for academic argumentation. For instance, statements such as "The 75th year of the State of Israel is no celebration" (referenced in row 15), "The current government, led by the national-religious right wing for the first time", "The emergence of a feeling of weakness has given rise to a fervent lust for power, transforming into killing machines, and in the ensuing rage unleashed upon Gaza, Israel is depicted as a warlord", and "the notion that the Jewish state is established through human sacrifice" are primarily subjective in nature. While these assertions might hold validity from an individual perspective, they lack the foundation in empirical evidence and rigorous analysis that is characteristic of academic discourse. Furthermore, it is imperative to acknowledge the highly sensitive and complex humanitarian issues in Gaza. However, the arguments presented by the author do not adhere to the standards of academic scrutiny and rigor, for more reasons:

The manuscript exhibits a noticeable absence of methodological clarity. The author fails to elucidate the methodologies employed in the study, leading to a predominantly descriptive, rather than analytical or critical, presentation of the scholars discussed. A critical aspect in academic writing is the explicit articulation of methods and materials used, including a rationale for the selection of specific scholars for analysis. This is particularly pertinent when some referenced texts do not directly address the contemporary state of Israel, as noted in lines 169-170 and 52-53.

Furthermore, the paper is characterized by assertions that are not substantiated with rigorous academic evidence. For example, the claim regarding the rabbinic tradition serving as a basis for racism in Israel (line 163) lacks empirical support. The reliance on secondary sources and references to scholars such as Schmitt, Walter Benjamin, Michel Foucault, Giorgio Agamben, and Derrida, without direct engagement with their primary works, further undermines the academic rigor of the paper. A critical deficiency in the paper is the lack of references for certain quoted authors and arguments.

Instances such as the mention of a Basic Law in 2018, Hermann Cohen’s interpretation of rationalization, and references to Yosef Haim Yerushalmi and Yirmiyahu Yovel (lines 137-138, 186, 188, and 224) are presented without appropriate citations. This absence of references detracts from the scholarly integrity of the work. Moreover, the paper does not provide working definitions for key concepts such as "Jewish State", "the Jewishness of the State of Israel", "secularism", "State of Israel", and "national-religious State". The lack of clear definitions and engagement with opposing viewpoints on these concepts represents a significant gap in the scholarly foundation of the paper. Additionally, while the paper describes important scholars relevant to the topic, it lacks a critical perspective on their works. The author’s approach is largely descriptive, with lines 149-158 raising numerous questions without offering corresponding analysis or answers.

Lastly, the paper includes references but omits a comprehensive bibliography, a fundamental component of academic writing that ensures transparency and allows readers to verify sources and further explore the topic.

Author Response

Thank you for the thoughtful review. My response:

  1. As to personal and subjective positions: I do not believe any research is conducted without a personal position. The question is how the methodology accounts for this position. I believe personal positions should be clearly stated in the introduction and in the conclusion so the reader is duely aware of the framing. This is what I did.
  2. As to methodology, it is once again clearly stated in the introduction: analysis of recent books by Jewish authors on Jewish statelessness. The article then proceeds to do just this.
  3. As to references, the authors metioned by the reviewer are discused by the authors that my paper reviewes. Since my methodology, as I explained it, focuses on the reviewed authors, I did not find it necessary to engage with the authors that they discuss. In some exceptions (Carl Schmitt) I did engage and referenced.
  4. As to the descriptive/critical stance, my understanding is that the paper does both. It provides concise presentations of the arguments presente dby the reviewed authors, and then proposes critical reflections on their work.
  5. Bibliography: indeed, necessary - I added a bibliography.

Reviewer 3 Report

Comments and Suggestions for Authors

Thank you very much.

‘Back to Exile: Current Jewish Critiques of the Jewish State’

Review comments:

Perhaps a less polemic opening would allow for the reader to enter the article with a more balanced state of mind; my fear is that statements like ‘while causing unprecedented harm to other values…and to the stability of the state’s institutions’ without further delineation may alienate some of the very readers who might most benefit from a broader consideration of the issues your paper covers. (Incidentally, there are strong arguments in favor of the legal proposals to adjust the power of the courts, particularly in the absence of a fully codified constitution. This is not to state that I support such – I feel unqualified to express either yea or nay – merely to point out the existence of these lines of thought.) Perhaps too ‘civil war’ is a bit alarmist as to my knowledge there was never a hint of protest groups actively taking up arms to attempt a forceful overthrow of the government.

The third paragraph is likewise too luridly written for an academic article. I would suggest reining in the ‘colorfulness’ of the language being used.

The first sentence in the fifth paragraph I think should read: ‘in the years before October 7’ as these citations are of course older. There are some other small typos throughout (nothing major but worth another proofread), and it might be good to uniform either using ‘he/his’ or ‘He/His’ when in reference to (the idea of) God.

The final line in the opening might be clearer if the questioned relationship between ‘secularism’ and ‘transcendence’ was put with a bit more detail; perhaps something along the lines of: ‘the question of secularism vis-à-vis presumptions of transcendence and conceptual relations to the people’; or the like.

What are the implications of Ophir not mentioning Islam? Does he take Islam to rightly present God as violence personified?

Out of curiosity, why not use ‘Torah’ instead of ‘Pentateuch’? If there is a reason, the reader may be interested in your choice (I am!).

I’m a little confused by the term ‘anarchic halakhah’; I know of many hard-to-rationalize halakhah, but not many that I might label as ‘anarchic’: could you please provide some examples?

How Sartrean is Trom?

The closing is powerful and provocative, but also – and perhaps purposefully – vague in conclusive offerings: does the author wish to remove the State of Israel as political entity in order to move her ‘into the Arab and Islamic space’; or is this a call for a re-invention of two states, or of a wholly other single state that is not ‘a Jewish state’ but is populated by its present citizenry? These questions pressed themselves upon me as reader, and it may be that the author wishes to leave them as open queries; I merely wish to share some reactions.

Thank you very much for a wonderfully interesting and engagingly written piece.

Author Response

Thank you for the thoughtful review, good suggestions and encouragement. My responses:

  1. As to the opening - I do not think it is so polemic, it is, like any other description, done from a certain personal perspective, but I find it pertinent to frame the paper through my own subjective position. As to the "colorfulness", this is the style I chose for this text.
  2. As for typos, I re-read and copyedited the text.
  3. As to Ophir and Islam - a good question, I'm not sure.
  4. Pentateuch - that is the term that Ophir used, he does not explain why not Torah.
  5. anarchic halakhah - means for Ophir a political system built not on a ruler (arche) but on a multitude of legal decisioners.
  6. Trom is not Sartrean since he does not at all reflect on any Jewish consciousness generated in response to anti-Semitism, or other
  7. Thank you for the questions in the end - I am not sure. In this context I preferred to leave open the precise political forms that should be adopted in order to achieve the stated goals.